# Performance and Pacing of Age Groups in Half-Marathon and Marathon

**DOI:** 10.3390/ijerph16101777

**Published:** 2019-05-20

**Authors:** Pantelis Theodoros Nikolaidis, Ivan Cuk, Thomas Rosemann, Beat Knechtle

**Affiliations:** 1Exercise Physiology Laboratory, 18450 Nikaia, Greece; pademil@hotmail.com; 2School of Health and Caring Sciences, University of West Attica, 12243 Egaleo, Greece; 3Faculty of Physical Education and Sports Management, Singidunum University, 1100 Belgrade, Serbia; ivan_cuk84@yahoo.com; 4Institute of Primary Care, University of Zurich, 8091 Zurich, Switzerland; thomas.rosemann@usz.ch; 5Medbase St. Gallen Am Vadianplatz, 9001 St. Gallen, Switzerland

**Keywords:** aging, endurance, gender, master athlete, performance

## Abstract

The aim of the present study was to examine the age-related differences in performance and pacing in a half-marathon compared to a marathon. All finishers (*n =* 9137) in the Ljubljana 2017 half-marathon (*n* = 7258) and marathon (*n =* 1853) with available data on split times during the races, were analysed for pacing. Half-marathoners were slower than marathoners among women, (2.77 ± 0.35 versus 2.86 ± 0.39 m·s^−1^ respectively, *p* < 0.001), but faster among men (3.14 ± 0.45 versus 3.08 ± 0.46 m·s^−1^ respectively, *p* < 0.001). In both race distances, the <25 age group was the fastest and the >54 age group the slowest (*p* < 0.001). All age groups presented a positive pacing in both race distances and genders, with each segment being slower than the previous one. However, an end spurt was observed in the marathon, but not in the half-marathon. A more even pace in the half-marathon than in the marathon was shown for most age groups. In summary, age-group finishers in the half-marathon decreased running speed across the race, presented a more even pacing than marathoners, and did not show an end spurt.

## 1. Introduction

Endurance exercise has been shown to play a beneficial role for health, e.g., it has been suggested that it reduces the risk of cardiovascular disease, stroke, diabetes, several cancers, depression, and falls [1]. This beneficial role of endurance exercise might explain the increased participation, especially of older age groups, in endurance races [2,3], and the increased scientific interest in studying participation and performance characteristics of master athletes [4]. Considering endurance running races, the half-marathon has been a race distance of increasing popularity. For instance, the number of finishers in the half-marathon in Switzerland increased from the year 2000 to 2010, by 299% in women and 231% in men [3]. Compared to the marathon, the number of annual races and finishers in the USA in the half-marathon was three and four times higher, respectively. [5]. So far, several aspects of the half-marathon have been studied, e.g., rates of participation, trends in performance, gender differences [6], perception of effort [7], heart rate response [8], biomechanics [9], and pacing of recreational [10] and elite runners [11]. However, no information was available on the pacing of age groups of half-marathon runners. Such information would be of great practical interest, since half-marathon runners compete usually in five-year age groups [3].

The variation of pacing by age group has been well-studied in marathon runners [12,13,14,15,16,17]. Older runners adopted a more even pacing (defined as the quotient of the speed in the last 9.7 km divided by the first 32.5 km) in a Midwestern USA marathon [12]. The relationship of pacing (considered either as the percentage difference between the fastest and slowest split, or as the percentage change of speed in consecutive splits) with age was also identified in two analyses of the New York City marathon, showing a more even pacing in the older runners [13,14]. This trend was also observed in the Chicago marathon [17], 14 USA marathons [16], and in a recent study in the Athens marathon [15]. The above-mentioned studies have indicated that the older runners presented less variation in their speed compared to their younger counterparts, as well as that women had more even pacing than men.

Although these studies [12,13,14,15] have improved our understanding of age- and gender-related differences in pacing in endurance running, they were conducted in marathons and their findings might not be “transferred” to the half-marathon. Compared to the marathon, the half-marathon has been a more “massive” sport event [5], presenting slower race speed [18] and inducing smaller muscle fatigue, muscle fiber damage, perceived muscle pain [19], and inflammation [20]. Since it has been suggested that age-related decrease in performance is dependent on race duration [21,22], it would be reasonable to assume that age-related differences in pacing might also vary by race distance. The knowledge about potential differences among age groups would be of great practical value for the sports medicine team (e.g., fitness trainers, exercise physiologist, coach) working with master endurance runners. Therefore, the aim of the present study was to examine age-related differences in performance and pacing in the half-marathon and the marathon.

## 2. Materials and Methods

### 2.1. Participants and Data Acquisition

This study was approved by the Institutional Review Board of Kanton St. Gallen, Switzerland, with a waiver of the requirement for informed consent of the participants as the study involved the analysis of publicly available data. The study was conducted in accordance with recognized ethical standards, according to the Declaration of Helsinki adopted in 1964 and revised in 2013. For the purpose of this study, we have included official results and split times from the publicly available “Ljubljana Marathon” website (http://vw-ljubljanskimaraton.si). Participants who did not finish the race, or did not have a record of any of the split times were excluded from the study. Finally, 1853 finishers of the 2017 Ljubljana marathon, and 7258 finishers of the 2017 Ljubljana half-marathon (total 9137 finishers) were included in this study. We chose the “Ljubljana Marathon” in particular, since both the marathon and the half-marathon were held on the same day and on the same track, thus eliminating the potential influence of the environmental conditions. Moreover, the half marathon race was entirely contained within the marathon race.

Both the marathon and the half-marathon were considered flat, with the elevation difference of only 29 m (ranging from 295 m to 324 m). For comparison, the Berlin marathon has an ascent of 21 m and a course record for men of 2:01:39, h:min:s, set in 2018 by Eliud Kipchoge [23], and the New York City marathon has a total ascent of 390 m with a course record of 2:05:06, h:min:s, set in 2011 by Geoffrey Kiprono Mutai [24]. The temperature ranged from 4.2 °C to 15.4 °C on the race day, without excess humidity or strong wind.

### 2.2. Data Analysis

In the first step of data analysis, we have calculated the mean speed for the entire race for each participant in the half-marathon and the marathon. Additionally, we have calculated the mean running speed in five race segments, for both the marathon and the half-marathon [10] that correspond to:Segment 1—Mean running speed from 0–23.7% of the race (0–5 km for the half-marathon and 0–10 km for the marathon)Segment 2—Mean running speed from 23.7–47.4% of the race (5–10 km for the half–marathon and 10–20 km for the marathon)Segment 3—Mean running speed from 47.4–71.1% of the race (10–15 km for the half–marathon and 20–30 km for the marathon)Segment 4—Mean running speed from 71.1–94.8% of the race (15–20 km for the half–marathon and 30–40 km for the marathon)Segment 5—Mean running speed from 94.8–100% of the race (20–21.0975 km for the half- marathon and 40–42.195 km for the marathon)Mean race speed—Mean running race speed 0–100% (0–21.0975 km for the half–marathon and 0–42.195 km for the marathon)

The aforementioned segments were subsequently expressed as a percentage faster or slower than the mean segment speed. The fastest segment for each individual was then named the “positive range,” (PR), while the slowest segment was named the “negative range” (NR). The absolute sum of the positive range and negative range was then calculated and named the “pace range”. This allowed for normalized speed comparisons between all athletes, as well as between the marathon and the half-marathon. Finally, to examine the final 2.195 km for the marathon and 1.0975 km for the half-marathon, the end spurt was defined as when the speed at segment 5 was faster than that at segment 4.

### 2.3. Statistical Analysis

To test differences in the pace range between marathon and half-marathon runners in eight age groups, two two-way analyses of variance (ANOVA) were performed (separately for men and women). Main effects of race (marathon and half-marathon), age group (<24; 25–29; 30–34; 35–39; 40–44; 45–49; 50–54; 55+), and their interaction (race × age group) were performed. An additional two two-way analyses of variance (ANOVA) were performed (separately for men and women) to test differences in pacing between marathon and half-marathon runners in nine age groups. Main effects of race (marathon and half-marathon), age group (<24; 25–29; 30–34; 35–39; 40–44; 45–49; 50–54; 55+), and their interaction (race × age group) were performed. For all ANOVAs, a Bonferroni post-hoc test was performed. Effect size was presented via eta squared (η^2^), where the values of 0.01, 0.06, and above 0.14 were considered small, medium, and large, respectively [25]. The men-to-women ratio (MWR) was calculated as the quotient of men divided by women finishers, and was used to describe the variation of gender participation by race distance and age group. A chi-square (χ^2^) examined the association of gender with race distance and age group, and Cramer’s phi (φ) evaluated the magnitude of these associations. Alpha level was set at *p* < 0.05. All statistical tests were performed using Microsoft Office Excel 2007 (Microsoft Corporation, Redmond, WA, USA) and SPSS 20 (IBM, Armonk, NY, USA).

## 3. Results

### 3.1. Participation by Gender, Race Distance, and Age Group

The number of men and women in each race distance and age group are presented in Table 1. The total MWR of finishers in both race distances was 1.82. A gender × race distance association on participation was observed (χ^2^ = 234.37, *p* < 0.001, φ = 0.16), with the MWR being higher in the marathon (3.94) than in the half-marathon (1.54). In the half-marathon, a gender × age group association in participation was shown (χ^2^ = 41.66, *p* < 0.001, φ = 0.08), with the largest MWR being in the >54 age group (2.42) and the lowest in the age group 25–29 years (1.32). In the marathon, a gender × age group association in participation was found (χ^2^ = 29.98, *p* < 0.001, φ = 0.13) with the largest MWR being in the age group >54 years (14.42) and the lowest in the age group <25 years (2.35).

### 3.2. Running Speed by Gender and Race Distance

A small main effect of gender on running speed was observed (*p* < 0.001, η^2^ = 0.052), with men (3.12 ± 0.45 m·s^−1^) being faster than women (2.78 ± 0.36 m·s^−1^) by 12.2% (Figure 1). No main effect of race distance on running speed was shown (*p* = 0.348, η^2^ < 0.001). A trivial gender × race distance interaction on race speed was found (*p* < 0.001, η^2^ = 0.004), with gender difference being higher in the half-marathon (+13.4%) than in the marathon (+7.6%). Half-marathoners were slower than marathoners among women (2.77 ± 0.35 versus 2.86 ± 0.39 m·s^−1^ respectively, *p* < 0.001), but faster among men (3.14 ± 0.45 versus 3.08 ± 0.46 m·s^−1^ respectively, *p* < 0.001).

### 3.3. Age by Gender and Race Distance

A trivial main effect of gender on age was observed (*p* < 0.001, η^2^ = 0.003), with men (41.2 ± 10.7 years) being older than women (39.5 ± 10.0 years) by 4.2%. A trivial main effect of race distance on age was shown (*p* = 0.004, η^2^ = 0.001), with marathon runners (41.7 ± 9.8 years) being older than their half-marathon peers (40.3 ± 10.7 years) by 3.5%. No gender × race distance interaction on age was found (*p* = 0.268, η^2^ < 0.001). Marathon runners were older than half-marathon runners in men (*p* < 0.001), but not in women (*p* = 0.297).

### 3.4. Race Speed by Age Group

In the half-marathon, a small main effect of age group on running speed (*p* < 0.001, η^2^ = 0.018) was observed for all finishers, with the age group <25 years being the fastest (3.08 ± 0.48 m·s^−1^), and the age group >54 years being the slowest (2.89 ± 0.42 m·s^−1^). The gender difference ranged from +12.2% (age group 30–34 years) to +15.7% (age group 25–29 years), however, no gender × age group interaction on running speed was shown (p =0.112, η^2^ = 0.002). In the marathon, a small main effect of age group on running speed (*p* < 0.001, η^2^ = 0.023) was found for all finishers, with the age group <25 years being the fastest (3.19 ± 0.68 m·s^−1^), and the age group >54 years being the slowest (2.87 ± 0.37 m·s^−1^). The gender difference ranged from +3.3% (age group >55 years) to +11.4% (age group 45–49 years). Nevertheless, no gender × age group interaction on running speed was observed (*p =* 0.525, η^2^ = 0.003).

A small main effect of age group on running speed was observed in women half-marathon runners (*p* < 0.001, η^2^ = 0.022), with the age group <25 years being the fastest (2.83 ± 0.36 m·s^−1^), and the age group >54 years being the slowest (2.63 ± 0.36 m·s^−1^). A main effect of age group on running speed was also shown in men half-marathon runners (*p* < 0.001, η^2^ = 0.023), with the age group <25 years being the fastest (3.25 ± 0.47 m·s^−1^), and the age group >54 years being the slowest (2.99 ± 0.40 m·s^−1^). A moderate main effect of age group on running speed was found in women marathon runners (*p* < 0.001, η^2^ = 0.075), with the age group 25–29 years being the fastest (3.10 ± 0.60 m·s^−1^), and the age group 50–54 years being the slowest (2.74 ± 0.29 m·s^−1^). A main effect of age group on running speed was also observed in men marathon runners (*p* < 0.001, η^2^ = 0.039), with the age group <25 years being the fastest (3.23 ± 0.70 m·s^−1^), and the age group >54 years being the slowest (2.88 ± 0.37 m·s^−1^).

### 3.5. Running Speed by Segment

The average running speeds for four segments, as well as end spurt, are presented in Table 2. From the descriptive data in Table 2, we can observe a gradual decrease in average speed through the race segments for both genders, in both the marathon and the half-marathon, and for all age groups with a characteristic end spurt. Moreover, we can observe the absence of an end spurt in half-marathon runners.

When pace range in men runners was observed, results showed significant main effects of race (smaller pace range in half-marathon than marathon, η^2^ = 0.12, *p* < 0.01), age group (η^2^ < 0.01, *p* < 0.01), and race × age group interaction (η^2^ < 0.01, *p* < 0.01). Within each race distance, a main effect of age group on pacing was observed for the half-marathon (*p =* 0.022, η^2^ = 0.004) and for the marathon (*p =* 0.031, η^2^ = 0.010), however, post-hoc comparisons did not reveal any significant difference.

When pace range in women runners was considered, the results showed significant main effects of race (smaller pace range in the half-marathon than the marathon, η^2^ = 0.02, *p* < 0.01), and age group (η^2^ = 0.01, *p =* 0.01). A significant main effect of the race × age group interaction was not obtained (η^2^ < 0.01, *p =* 0.07). Within each race distance, a main effect of age group on pacing was shown in the half-marathon (*p* < 0.001, η^2^ = 0.015), but not in the marathon (*p =* 0.357, η^2^ = 0.021). In the half-marathon, the age group 30–34 years had less pace range than the age group <25 years (−2.1%), the age group 25–29 years (−1.9%), the age group 50–54 years (−2.4%), and the age group >54 years (−3.2%). The age group >54 years had a larger pace range than the age group 35–39 years (+1.8%), the age group 40–44 years (+2.2%), and the age group 45–49 years (+1.8%).

With regards to the appearance of an end spurt (i.e., faster at segment 5 than at segment 4), an end spurt × race distance association was observed (χ^2^ = 1065.2, *p* < 0.001, φ = 0.342), with more marathon runners (71.1%) showing an end spurt than half-marathon runners (29.8%) (Figure 2). In the half-marathon, an end spurt × race distance association was shown (χ^2^ = 37.2, *p* < 0.001, φ = 0.072), with more women (33.9%) presenting an end spurt than men (27.2%). In the marathon, an end spurt × race distance association was found (χ^2^ = 41.4, *p* < 0.001, φ = 0.150), with more women (84.5%) presenting an end spurt than men (67.7%). In the half-marathon, an end spurt × age group association was observed in women (χ^2^ = 31.49, *p* < 0.001, φ = 0.105) and in men (χ^2^ = 122.0, *p* < 0.001, φ = 0.160). In the marathon, no end spurt × age group association was observed in women (χ^2^ = 5.98, *p =* 0.542, φ = 0.126) or in men (χ^2^ = 4.60, *p =* 0.709, φ = 0.056).

## 4. Discussion

The main findings of the present study were that: (a) the MWR was larger in the marathon than in the half-marathon and in the older than in the younger age groups. (b) Men were both faster and older than women with the gender difference in performance being larger in the half-marathon than in the marathon. (c) In both genders and distances, younger age groups were faster than their older peers. (d) All age groups presented a positive pacing in both race distances and genders, with each segment being slower than the previous one. (e) All age groups in both genders showed an end spurt in the marathon, but not in the half-marathon. (f) Most age groups in both genders exhibited a more even pace in the half-marathon than in the marathon.

### 4.1. Men-to-Women Ratio by Race Distance and Age Group

The smaller MWR in the half-marathon than the marathon indicated that a relatively (%) larger number of women participated in the former than in the latter. This finding confirmed previous research analysing all half-marathons and marathons in Switzerland from the year 2000 to 2010, showing that a relatively larger number of women participated in half-marathons than in marathons [26]. In addition, the larger MWR in the older age groups than among their younger peers suggested a relatively (%) smaller participation of women in older age groups. This observation was in agreement with a recent study in GöteborgsVarvet, the world’s largest half-marathon, where MWR in finishers increased with age [6]. An explanation of the relatively small number of women finishers in older age groups might be their historically later engagement in endurance running races, however, it should be highlighted that there is a trend of decreasing MWR in endurance races across calendar years [27].

### 4.2. Performance

The faster race speed in men than in women was in line with previous observations in endurance running races [6] and the gender differences in physiological correlates of performance, such as maximal oxygen uptake [28]. The younger age of women than men might explain the relatively smaller number of women in the older age groups. The faster performance of younger age groups than their older counterparts in the half-marathon was a novel finding, and confirmed age-related differences reported previously for the marathon [29]. It should be highlighted that the magnitude of the age-related differences was smaller in the half-marathon than in the marathon, indicating that aging played a less important role in the former than in the latter race distance.

### 4.3. Pacing and End Spurt

Compared to the marathon, the half-marathon had two unique characteristics. First, there was no end spurt of age groups in this race, and second, age groups had a more even pacing. It should be highlighted that these two characteristics were inter-related, i.e., an absence of an end spurt was accompanied by a more even pacing, and vice versa. The occurrence of an end spurt in the marathon was in agreement with previous findings in marathon age groups, e.g., finishers in the New York City marathon [14]. The absence of an end spurt in the half-marathon might be attributed to a more ‘all-out’ effort as a shorter distance than the marathon. Both race distances showed a positive pacing, i.e., running speed decreased across the race. The relatively more even pacing in the half-marathon than the marathon might likely be due to the occurrence of less fatigue in the former race [19]. Marathon runners were able to maintain a similar average running speed to half-marathon runners, despite running double the distance. With regards to the age-related differences in pacing, significant differences were observed in women half-marathon runners only, and did not indicate a specific trend across the life-span. These findings were in disagreement with the existing literature on marathon races. For instance, the analysis of the New York City Marathon [30], a mid-western USA marathon [12], a Chicago marathon [17], and 14 USA marathons [16], indicated a more even pacing (i.e., smaller pace range) in the older age groups. This disagreement might be partially attributed to the relatively small sample of women half-marathon (*n =* 2852) and marathon runners (*n =* 375) in the present study compared to the above-mentioned studies in USA marathons (e.g., the analysis of the New York City Marathon included 117,595 women [30]). Nevertheless, the lack of an end spurt in older half-marathon runners might indicate that this age group adopted a pacing with less variation than their younger peers.

Overall, age did not play an important role on the pacing of finishers in both the half-marathon and the marathon. This trend might be due to the beneficial role of training that has been observed, even for older age groups. For instance, it has been shown that a higher volume of training was associated with favorable inflammatory and redox profiles at rest in older marathon runners [31]. Moreover, it has been proposed that some lifestyle and behaviour factors might be more important than age in relation to marathon performance, and that many master marathon finishers were relatively ‘new’ to running, having been engaged in regular training for just a few years [32]. Furthermore, exercise might attenuate the loss of skeletal muscle mass and function to prevent muscle aging comorbidities, and to improve physical performance and quality of life [33]. In advanced age, physical activity might mitigate sarcopenia, restore robustness, and counterbalance the development of disability [1]. Thus, master endurance runners might present similar performance aspects, e.g., pacing, compared to their younger peers due to their regular exercise.

### 4.4. Limitations, Strength, and Practical Applications

A limitation of the present study was the specific characteristics of this race, which could be classified as an ‘average’ race with regards to its rates of participations. Thus, caution would be needed to generalize its findings into other half-marathons. Moreover, the role of environmental conditions was not considered, since detailed data on temperature, humidity, and wind were not available, and these environmental parameters might influence endurance performance [34]. For instance, it has been observed recently that an increase in temperature by 1 °C was related to a slower race time by ~2 min in the Boston marathon [35]. To study the effect of environmental conditions in the Ljubljana half-marathon and marathon in the future, it would be necessary to consider a large number of calendar years following the paradigm of the Boston marathon [34,35]. On the other hand, a strength of the study was its novelty, since it was the first to examine performance and pacing in age groups of half-marathon runners. The findings were of both theoretical and practical interest. From a theoretical point of view, exercise physiologists and gerontologists interested in examining the master athlete as a model of healthy aging, would benefit from the information about performance and pacing trends in the older age groups. From a practical perspective, coaches and fitness trainers might work with endurance runners, differing in age and participating in both half-marathon and marathon races.

## 5. Conclusions

In summary, age-group finishers in the half-marathon decreased speed across the race, presented a more even pacing than marathoners, and did not show an end spurt. As this was the first study on performance and pacing of age-group finishers in the half-marathon, the findings had practical applications for a large number of coaches and fitness trainers working with endurance runners.

## Figures and Tables

**Figure 1 ijerph-16-01777-f001:**
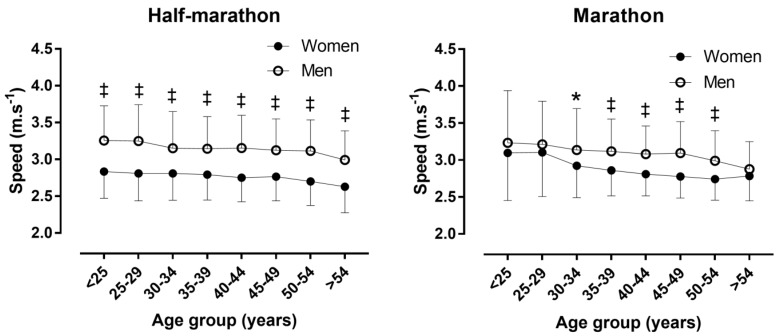
Race speed by race distance, gender, and age group. Error bars represent standard deviations, * *p* < 0.05; ^‡^
*p* < 0.001.

**Figure 2 ijerph-16-01777-f002:**
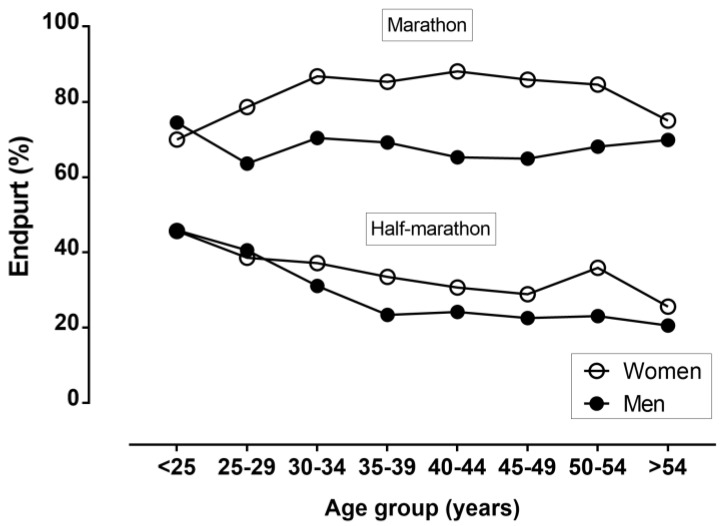
Variation of end spurt by gender, race distance, and age groups.

**Table 1 ijerph-16-01777-t001:** Distribution of men and women in each race and age group.

Age Groups	Men			Women		
	Marathon	Half-Marathon	Total	Marathon	Half-Marathon	Total
<25	47	268	315	20	199	219
25–29	99	433	532	28	329	357
30–34	179	589	768	53	393	446
35–39	292	753	1045	68	504	572
40–44	297	785	1082	84	540	624
45–49	231	646	877	71	436	507
50–54	160	441	601	39	248	287
>54	173	491	664	12	203	215
Total	1478	4406	5884	375	2852	3227

**Table 2 ijerph-16-01777-t002:** Segments and race speed for men and women, marathon and half marathon runners for each age group.

		Men				Women			
		Marathon		Half-Marathon		Marathon		Half-Marathon	
		Mean	SD	Mean	SD	Mean	SD	Mean	SD
Age 18–24	Segment 1	3.43	0.67	3.37	0.48	3.29	0.61	2.97	0.34
	Segment 2	3.37	0.70	3.33	0.47	3.22	0.62	2.90	0.36
	Segment 3	3.26	0.75	3.21	0.49	3.08	0.69	2.77	0.38
	Segment 4	2.99	0.77	3.17	0.51	2.88	0.69	2.76	0.42
	Segment 5	3.10	0.64	3.18	0.51	2.97	0.61	2.76	0.41
Age 25–29	Segment 1	3.44	0.57	3.34	0.49	3.25	0.59	2.95	0.34
	Segment 2	3.38	0.57	3.32	0.48	3.19	0.60	2.88	0.37
	Segment 3	3.23	0.61	3.21	0.50	3.09	0.64	2.74	0.39
	Segment 4	2.94	0.64	3.18	0.54	2.93	0.62	2.72	0.42
	Segment 5	3.02	0.60	3.14	0.55	3.03	0.53	2.70	0.41
Age 30–34	Segment 1	3.35	0.53	3.27	0.49	3.11	0.40	2.92	0.35
	Segment 2	3.29	0.54	3.23	0.49	3.02	0.43	2.87	0.36
	Segment 3	3.16	0.59	3.10	0.51	2.90	0.46	2.75	0.38
	Segment 4	2.86	0.63	3.07	0.54	2.73	0.49	2.75	0.40
	Segment 5	2.97	0.58	3.01	0.55	2.90	0.45	2.72	0.40
Age 35–39	Segment 1	3.34	0.42	3.27	0.43	3.02	0.33	2.93	0.32
	Segment 2	3.28	0.42	3.22	0.43	2.93	0.35	2.86	0.34
	Segment 3	3.13	0.47	3.10	0.44	2.84	0.37	2.72	0.36
	Segment 4	2.85	0.50	3.06	0.47	2.70	0.39	2.72	0.38
	Segment 5	2.94	0.46	2.98	0.48	2.82	0.40	2.69	0.38
Age 40–44	Segment 1	3.27	0.37	3.28	0.43	2.98	0.25	2.89	0.30
	Segment 2	3.20	0.37	3.23	0.43	2.89	0.27	2.81	0.33
	Segment 3	3.09	0.40	3.10	0.46	2.76	0.32	2.68	0.34
	Segment 4	2.85	0.43	3.07	0.49	2.65	0.36	2.68	0.36
	Segment 5	2.92	0.44	2.99	0.49	2.79	0.35	2.64	0.36
Age 45–49	Segment 1	3.31	0.41	3.24	0.41	2.97	0.26	2.91	0.30
	Segment 2	3.24	0.41	3.20	0.42	2.88	0.28	2.83	0.33
	Segment 3	3.11	0.45	3.07	0.43	2.73	0.34	2.69	0.34
	Segment 4	2.83	0.48	3.04	0.47	2.59	0.35	2.68	0.36
	Segment 5	2.90	0.46	2.96	0.47	2.72	0.33	2.64	0.36
Age 50–54	Segment 1	3.24	0.38	3.23	0.41	2.95	0.25	2.86	0.30
	Segment 2	3.14	0.39	3.19	0.42	2.84	0.27	2.77	0.32
	Segment 3	2.98	0.43	3.07	0.43	2.68	0.34	2.62	0.34
	Segment 4	2.72	0.47	3.04	0.46	2.57	0.33	2.61	0.37
	Segment 5	2.82	0.43	2.96	0.47	2.67	0.32	2.58	0.37
Age 55+	Segment 1	3.10	0.35	3.13	0.38	2.99	0.30	2.80	0.31
	Segment 2	3.01	0.37	3.07	0.39	2.90	0.29	2.70	0.36
	Segment 3	2.86	0.41	2.93	0.41	2.75	0.36	2.55	0.37
	Segment 4	2.64	0.42	2.90	0.43	2.58	0.41	2.53	0.39
	Segment 5	2.73	0.42	2.81	0.44	2.69	0.40	2.47	0.39

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
