# Peer review of "Performance and Pacing of Age Groups in Half-Marathon and Marathon"

_ijerph, 2019, doi:10.3390/ijerph16101777_

Round 1

Reviewer 1 Report

The introduction does not offer a convincing rationale for the study. In particular, the opening paragraph which focuses on the potential benefits of exercise in an elderly population does not align with the research question or analyses that have been undertaken especially since the oldest age category analysed is >54. Previous literature has demonstrated that age-related performance declines in marathon and half marathon is not observed until  >54 (Leyk et al 2007) or >60 years (Knechtle and Nikolaidis 2018) therefore by having all of the older age groups in one category means that important findings in these  older age groups has been missed. Furthermore age groups <54 are not  ‘elderly’ so the rationale does not work. Thirdly, the authors suggest that findings of previous literature from marathons and other sports may not ‘transfer’ to half marathon but offer no convincing reason as to why this might be the case. Is there any evidence from  literature e.g. Hanley’s 2014 study on pacing of the IAAF world half-marathon championships or physiological/psychological demands of half marathon that might suggest the pacing of half marathon may be different to the marathon?

Approach to analysis of the data and statistical analysis is appropriate and generally consistent with previous similar studies on pacing. However the analyses undertaken for participation data are not described in the statistical analysis section.

Please be careful that terms are expressed in full before abbreviations are presented e.g. ES and MWR on page 3.

Results are extensive and it takes a good while to get to the data that actually reflects the research question. Is there any benefit in including Table 2, do Figures 2 and 3 offer much clearer and concise data on pace?

There are no references in the paragraph 4.2 performance yet the text includes statements where findings are in line with previous observations or confirm previous observations – these need to be referenced.

Should the first part of 4.3 where suggestions for lack of end spurt in half marathon are linked to ‘all out’ efforts relate back to Figures 2 and 3 where the Pace range is presented to confirm claims of more even paced approaches?

Do the authors have any suggestions as to why their data on women’s pacing is different to the previous literature identified in this section?

The final paragraph of section 4.3 links back to health benefits of exercise in older individuals – see previous comments on age categories. Furthermore, Leyk et al (2009) suggest that some lifestyle and behaviour factors may be more important than age in relation to marathon performance and that many older marathon finishers are relatively ‘new’ to running having engaged in regular running for just a few years.

Author Response

Thank you for your work and time on this review.

Comments and Suggestions for Authors

The introduction does not offer a convincing rationale for the study. In particular, the opening paragraph which focuses on the potential benefits of exercise in an elderly population does not align with the research question or analyses that have been undertaken especially since the oldest age category analysed is >54. Previous literature has demonstrated that age-related performance declines in marathon and half marathon is not observed until  >54 (Leyk et al 2007) or >60 years (Knechtle and Nikolaidis 2018) therefore by having all of the older age groups in one category means that important findings in these  older age groups has been missed. Furthermore age groups <54 are not  ‘elderly’ so the rationale does not work. Thirdly, the authors suggest that findings of previous literature from marathons and other sports may not ‘transfer’ to half marathon but offer no convincing reason as to why this might be the case. Is there any evidence from  literature e.g. Hanley’s 2014 study on pacing of the IAAF world half-marathon championships or physiological/psychological demands of half marathon that might suggest the pacing of half marathon may be different to the marathon?

Answer: We agree with the expert reviewer and revised the introduction as suggested; we turned the focus from the elderly to the pacing in half-marathon by age groups, and developed the unique characteristics of half-marathon. Accordingly, we rewrote the largest part of the introduction.

Approach to analysis of the data and statistical analysis is appropriate and generally consistent with previous similar studies on pacing. However the analyses undertaken for participation data are not described in the statistical analysis section.

Answer: We agree with the expert reviewer and developed the participation aspect in the statistics section (“Men-to-women ratio (MWR) was calculated as the quotient of men divided by women finishers, and was used to describe the variation of gender participation by race distance and age group. A chi-square (χ2) examined the association of gender with race distance and age group, and Cramer’s phi (φ) evaluated the magnitude of these associations.”).

Please be careful that terms are expressed in full before abbreviations are presented e.g. ES and MWR on page 3.

Answer: We agree with the expert reviewer and corrected all text accordingly.

Results are extensive and it takes a good while to get to the data that actually reflects the research question. Is there any benefit in including Table 2, do Figures 2 and 3 offer much clearer and concise data on pace?

Answer: We agree with the expert reviewer and deleted figures 2 and 3 as data are better described in table 2. In addition, we eliminated the third decimal in speed (m/s) to read easier.

There are no references in the paragraph 4.2 performance yet the text includes statements where findings are in line with previous observations or confirm previous observations – these need to be referenced.

Answer: We agree with the expert reviewer and added three references.

Should the first part of 4.3 where suggestions for lack of end spurt in half marathon are linked to ‘all out’ efforts relate back to Figures 2 and 3 where the Pace range is presented to confirm claims of more even paced approaches?

Answer: We agree with the expert reviewer and reported this trend („It should be highlighted that these two characteristics were inter-related, i.e. an absence of end spurt was accompanied by a more even pacing and vice versa.“).

Do the authors have any suggestions as to why their data on women’s pacing is different to the previous literature identified in this section?

Answer: We agree with the expert reviewer and developed the interpretation of this aspect („This disagreement might be partially attributed to the relatively small sample of women half-marathon (n=2,852) and marathon runners (n=375) in the present study compared to the abovementioned studies in USA marathons (e.g. the analysis of the New York City Marathon included 117,595 women [27]).“).

The final paragraph of section 4.3 links back to health benefits of exercise in older individuals – see previous comments on age categories. Furthermore, Leyk et al (2009) suggest that some lifestyle and behaviour factors may be more important than age in relation to marathon performance and that many older marathon finishers are relatively ‘new’ to running having engaged in regular running for just a few years.

Answer: We agree with the expert reviewer and updated this section considering the revision of the introduction.

Reviewer 2 Report

Here are my considerations for authors:

Congratulations for the originality and quality of the study. In opinion of this reviewer, this paper deals with a very interesting topic, and the research design seems appropriate to answer the study aims. However, there are some formal aspects to be addressed:

Page 1, line 19 “Sexes”, change for “Genders”. Also “endspurt” should be separated (end spurt), both along the text.

Page 2, lines 20-21. Probably would need to address in the future the meteorological conditions of the race, mainly wind, as I assume is a crucial variable, more than there were no "strong wind", as this variable is key for race performance. Also, temperature and humidity are key. Probably, a better treatment of this to crucial variables should be addressed in the discussion part.

Page 2, lines 20-21.Also, slope is crucial, and you refer to this race as “considered as flat”. Authors should explain why is flat as there was a difference 29 m in height. Is there any previous reference dealing with slope along half/marathon races? If so, please discuss.

Page 3, line 15. “MWR” abbreviation has not been explained before, and it is crucial for results and discussion interpretation. Please explain.

Page 8, line 6. Use “neither” just after parenthesis instead “and”.

Related results, they are clear and well presented, but probably too much extensive and not all dealing with research question. Consider to reduce them (figures 2 and 3?)

Author Response

Comments and Suggestions for Authors

Here are my considerations for authors:

Congratulations for the originality and quality of the study. In opinion of this reviewer, this paper deals with a very interesting topic, and the research design seems appropriate to answer the study aims. However, there are some formal aspects to be addressed:

Page 1, line 19 “Sexes”, change for “Genders”. Also “endspurt” should be separated (end spurt), both along the text.
Answer: We agree with the expert reviewer and revised all text thoroughly according to these suggestions.

Page 2, lines 20-21. Probably would need to address in the future the meteorological conditions of the race, mainly wind, as I assume is a crucial variable, more than there were no "strong wind", as this variable is key for race performance. Also, temperature and humidity are key. Probably, a better treatment of this to crucial variables should be addressed in the discussion part.
Answer: We agree with the expert reviewer and added this aspect in the limitations section.

Page 2, lines 20-21.Also, slope is crucial, and you refer to this race as “considered as flat”. Authors should explain why is flat as there was a difference 29 m in height. Is there any previous reference dealing with slope along half/marathon races? If so, please discuss.
Answer: We agree with the expert reviewer and clarified this aspect. We added in that section ‘To compare: The ‘Berlin Marathon’ has an ascent of 21m (www.bmw-berlin-marathon.com/dein-rennen/start-strecke-ziel/strecke/) and a course record for men of 2:01:39 h:min: set in 2018 by Eliud Kipchoge. The ‘New York City Marathon’ has a total ascent of 390 m (http://help.tcsnycmarathon.org/customer/en/portal/articles/2053698-race-course) with a course record of 2:05:06 h:min:s set in 2011 by Geoffrey Kiprono Mutai’ to show the difference in altitude between a flat marathon (Berlin) and another marathon.

Page 3, line 15. “MWR” abbreviation has not been explained before, and it is crucial for results and discussion interpretation. Please explain.
Answer: We agree with the expert reviewer and explained it.

Page 8, line 6. Use “neither” just after parenthesis instead “and”.
Answer: We agree with the expert reviewer and corrected it as suggested.

Related results, they are clear and well presented, but probably too much extensive and not all dealing with research question. Consider to reduce them (figures 2 and 3?)

Answer: We agree with the expert reviewer and removed figures 2 and 3, and inserted some of their information within the text.